# The sexual and reproductive health needs and preferences of youths in sub-Saharan Africa: A meta-synthesis

**Victoria Kalu Uka**[1]*, **Helen White**[1], **Debbie M. Smith**[2]

1 Division of Nursing, Midwifery and Social Work, School of Health Sciences, Faculty of Biology Medicine and Heath, University of Manchester, Manchester, United Kingdom, 2 Division of Psychology and Mental Health and Manchester Centre for Health Psychology (MCHP), School of Health Sciences, Faculty of Biology Medicine and Heath, University of Manchester, Manchester, United Kingdom

* victoria.uka@postgrad.manchester.ac.uk

**Data Availability Statement:** All relevant data are within the manuscript and its Supporting information files.

**Funding:** The author(s) received no specific funding for this work.

## Abstract

The sexual and reproductive health (SRH) needs of youths in sub-Saharan Africa are not being fully met, as evidenced by high rates of unintended pregnancies and sexually transmitted infections in this population. Understanding service needs and preferences of sub-Saharan African youths aged 10–24 years is critical for improving access and SRH outcomes and the focus of this systematic review of qualitative research. Four databases were searched with key words to identify relevant studies, supplemented by citation search, with an update in June 2023. The eligibility criteria were clear and developed a priori. Twenty included studies from seven countries underwent quality appraisal using the Critical Appraisal Skills Programme (CASP). A meta-ethnographic approach was used to synthesise concepts across studies by the researchers. Four key themes were generated: *information needs; service needs; social needs; and delivery preferences*. Information needs encompassed desires for age-appropriate education on contraception, safer sex, bodily changes, and healthy relationships to fill knowledge gaps. Social needs consisted of life skills training, vocational development, substance use rehabilitation, and support systems to foster healthy behaviours. Service needs included accessible youth-friendly sexual health services, preventative care, sexually transmitted Infections (STI) management, and contraception; and delivery preferences including competent providers who maintain privacy and confidentiality, convenient youth-oriented settings, free or low-cost provisions, and youth involvement in service design. In conclusion, the identified themes emphasise the diverse nature of SRH needs and preferences among sub-Saharan African youths. Insights from their unique priorities and unmet needs inform policy development and intervention strategies. Tailored awareness campaigns, youth-centred training for providers, youth-friendly and confidential SRH models, comprehensive education, and engaging youth in developing relevant solutions may improve acceptability, access, and health outcomes. These efforts could address barriers around stigma, costs, and lack of knowledge, contributing to enhanced SRH and wellbeing. Fulfilling youth SRH needs in sub-Saharan Africa requires commitment across sectors to evidence-based, youth-focused strategies placing their perspectives at the centre.

**Competing interests:** the authors have declared that no competing interest exist.

## Introduction

According to World Health Organisation (WHO), youth are defined as those aged 10–24 years [1]. Youths make up a large portion of the population in sub-Saharan Africa (SSA) and are critical to nation-building [2, 3]. However, access to sexual and reproductive health services (SRHS) is restricted, due to religious, social, economic, and political barriers [4]. These barriers contribute to significant disparities in youth SRHS access and utilisation across SSA countries [5–8]. While political leaders in sub-Saharan Africa may embrace international support and funding for youth SRHS, actual implementation within these countries often face challenges [9–12]. Each year, over 374 million new cases of sexually transmitted infections (STIs) occur globally, predominantly among youth aged 15–24 years, with a significant proportion in low- and middle-income countries (LMICs) [13, 14]. In sub-Saharan Africa (SSA), youth aged 15–24 years account for a considerable portion of these cases, reflecting the high burden of STIs in this region [13]. Approximately 1.5 million HIV-positive youth reside in SSA [15]. Moreover, gender inequality in SSA exacerbates sexual health challenges, with many females experiencing unmet contraception needs due to intimate partner violence, which prevents them from negotiating safer sex [16]. Each year, 21 million pregnancies occur among 15-19-year-olds in LMICs, with half being unplanned [13, 14]. In SSA, unplanned pregnancies among adolescents contribute to increased female school dropout rates, further widening gender disparities in education [17, 18]. With abortion still illegal in some countries in SSA, including Nigeria, unintended pregnancy raises the vulnerability of young females to unsafe options and complications [19]. Such situations strain youths emotionally and psychologically [20], lowering self-esteem and disconnecting them from more empowered peers [21]. Healthy development is hindered [22] and potentially leads to substance misuse or mental health issues [23]. This connection between unmet sexual and reproductive health needs and hindered development is crucial to understand. As emphasised in [22], youth is a critical period for healthy development, including sexual development. When sexual and reproductive health needs are not adequately addressed, it can impede overall healthy development in youths, potentially leading to various negative outcomes. A lack of access to SRHS exacerbates these health problems [4], increasing adverse outcomes like illness and death [24].

When appropriate SRHS are available, including family planning and abortion services, youth vulnerabilities to unintended pregnancy and STIs can be reduced [25–28]. SRHS designed based on youth needs and preferences will promote usage [4]. The WHO, in collaboration with the Joint United Nations Programme on HIV/AIDS (UNAIDS) recommend youth participation in healthcare design and delivery, given their awareness of personal needs, to develop relevant solutions [14]. Ignoring youth perspectives risks service refusal and may stall progress towards quality sexual and reproductive health services [5, 14]. The current review explored the needs and preferences of youths of SRHS in SSA, with the aim to increase access.

## Methods

This review employed a systematic approach. The revised Preferred Reporting Items for Systematic Review and Meta-analysis (PRISMA) standards were followed to guide the identification and selection of relevant literature [29]. This review protocol was registered on PROSPERO (CRD42022307530).

### Ethics statement

As this study is a systematic review of published literature, it did not involve human or animal subjects, and therefore, did not require ethical approval. No primary data collection was conducted, and all data used were from publicly available sources.

## Search strategy

The review question and search terms were structured according to the Joanna Briggs Institute's (JBI) suggested framework: PICo, which represents Participants, Phenomena of interest, and Context [30] (S1 Fig). An initial scoping search was conducted in Ovid Medline to refine the review question and search terms. During the systematic search, medical subject headings (MeSH) were used in different databases to broaden the search terms [31]. The following four databases were searched: "Medical Literature Analysis and Retrieval System" (Ovid Medline), "Cumulative Index to Nursing and Allied Health Literature" (CINAHL plus), Psychological Information Database (PsycINFO), and Allied and Complementary Medicine Database (AMED). A citation search to identify additional relevant studies was also conducted. The S2 File presents the full search history on Ovid Medline, CINAHL plus and PsycINFO.

## Eligibility criteria

**Population.**   Youths aged 10–24 years in SSA were the target population. This is in line with the WHO's definition of youth [1]. Excluded groups were homeless, internally displaced persons, and refugees as their SRHS needs may differ substantially due to their situation [32, 33].

**Phenomena of interest.**   Studies must have reported at least one of these relevant phenomena regarding SRHS: needs, requirements, preferences, choices, options, wants, or desires. Studies relating to general healthcare needs rather than SRHS specifically were excluded.

**Context.**   All countries in SSA were included in accordance with review aims. Likewise, all SRHS locations were considered, including schools, marketplaces, transport terminals, clubhouses, community centres, health facilities [34]. Both facility-based and community-based SRHS settings were eligible.

**Publication types and study design.**   Inclusion was limited to published, peer-reviewed studies. Grey literature was excluded to ensure the quality and reliability of included sources, despite the potential for publication bias [35]. This decision was based on challenges in comprehensively identifying and accessing these sources, their heterogeneous nature, and potential lack of rigorous peer review. To mitigate potential bias, we conducted thorough searches across multiple major health and social science research databases, including medical, nursing, psychological, and allied and complementary medicine literature databases. Included papers were written in the English language, with the majority of African studies publish in English [36]. Unpublished studies, theses, conference papers, duplicates, and anecdotal reports were excluded because they may lack a transparent peer review process that ensures methodological rigor and may contain preliminary or duplicated data [35, 37]. Qualitative studies and mixed methods studies with distinct qualitative components (example, interviews, focus groups, observations) were included as they provide deeper understanding of youth's needs and preferences related to SRHS [38, 39].

## Study selection

Identified articles were transferred into EndNote version 20 and then into Rayyan software [40, 41] Besides manual removal, both software programs enabled the elimination of duplicates. Rayyan was used as a platform to screen article titles, abstracts, and full texts based on the eligibility criteria stated above to identify included reports ([41]; (See S2 Fig for PRISMA flow diagram). Prior to finalising the selection process, 10% check of titles and abstracts was conducted on the retrieved articles by two researchers independently (VU and LM) to compute an inter-rater reliability. We used Cohen's kappa to assess inter-rater reliability. The specific Kappa values and their implications are reported in the results section.

## Data extraction

A modified data extraction tool from the Cochrane Collaboration was used [42]. The tool allowed for information such as author names, publication date, study title, study question/ aims, participant demographics, design, population of interest, analysis methods, and findings to be captured from relevant studies. Data extraction was conducted independently by two researchers (VU and DS) with only qualitative data considered during the process.

## Quality appraisal

The Critical Appraisal Skills Programme (CASP) qualitative checklist from the Cochrane Qualitative and Implementation Group was adopted to judge the quality of the included studies [43]. Two researchers (VU and HW) independently assessed the quality of the included papers. The checklist consists of 10 questions (Q) focused on various aspects of the studies, including study aims, data collection methods, analysis, and results. Questions one to nine are closed-ended with three possible responses: 'yes,' 'can't tell,' and 'no,' while Q10 is an open-ended question. As noted by Long et al. [44], the CASP tool does not provide a standardised method for assessing the clarity and appropriateness of qualitative reporting. Therefore, to enhance transparency and facilitate inter-study comparison, we assigned scores to questions 1–9: 'yes' = 2, 'can't tell' = 1, and 'no' = 0. For Q10, we assigned scores of 2 for papers considered valuable and 1 for less valuable papers, based on our qualitative judgment (See S2 Table for quality appraisal summary). The scoring system allowed us to standardise the appraisal process, facilitate comparisons across studies, support overall quality judgments by categorising studies into low, medium, and high quality, and ensure transparency in our judgment. Importantly, no studies were excluded based on their scores. All papers, irrespective of quality scores, were included for their valuable contributions to the meta-ethnographic synthesis.

## Data synthesis

A meta-ethnographic approach, developed by Noblit, Hare and Hare [45], was employed to explore the findings of the primary studies. This method allows for comparative integration of data rather than just description and is thus considered appropriate [46, 47]. Taking a meta-ethnography approach systematically transforms findings by drawing comparisons through the process of translation [45, 48]. Consequently, this approach enabled synthesis of youth's SRHS needs and preferences across individual papers with greater explanatory power than narrative or thematic approaches [49, 50]. Its analytical nature and widespread use in non-ethnographies underscore its value [51–54].

   The included papers were read and re-read to identify codes and examine interrelationships [45]. Studies were then translated into one another through reciprocal translation to determine similarities and refutational synthesis to identify discrepancies between metaphors [45, 46, 55]. Subsequently, lines of argument were developed to reach a holistic interpretation [56]. In meta-ethnography, quality refers not to methods but to metaphor adequacy for rich data [48, 57]. To reduce bias, VU, DS, and HW were independently involved in synthesis of the data.

# Results

## Study selection

As illustrated in the PRISMA flow diagram (S2 Fig), an initial search was conducted in April 2022, and an updated search was performed in June 2023. The June 2023 search narrowed the date range to January 2022 to June 2023, aiming to capture papers published since the initial search conducted between February and April 2022. The combined searches yielded a total of

3,085 papers. After removing duplicates, 2,237 titles and abstracts were screened based on eligibility criteria. Of these, 2,216 papers were excluded for various reasons, including deficiency in more than one exclusion criterion (1,324), an incorrect target population (84), an incorrect phenomenon of interest (402), an unsuitable study design (95), either systematic or scoping reviews (60), grey literature (246), questionnaire development (1), and non-English language (1).

Cohen's kappa (κ) computation was 0.12, indicating slight agreement. This necessitated a review of the inclusion criteria, and more terms (wants and options) were introduced to represent the needs and preferences of youth and expand the phenomenon of interest component. Then, a second check was done since slight agreement may not allow confidence in the review process [58–60]. Cohen's value for the second 10% check revealed 0.66, indicating substantial agreement, and subsequently disagreements were settled by consensus [58, 61].

Twenty-four articles passed abstract screening, and attempts were made to retrieve full texts. However, one full-text article that passed initial screening based on its title and abstract could not be accessed despite multiple attempts. The article was not available through open access or institutional subscriptions, and no author contact information was available to request the full text. As a result, this article was excluded. Consequently, 23 full-text articles were reviewed, of which three were subsequently excluded because one lacked data from youths while the other two used quantitative methods for data analysis. This led to the inclusion of 20 studies in the final review. The findings from the extraction are presented in a summary table reflecting the extent of available evidence in the accepted studies (see S1 Table for data extraction summary).

## Quality appraisal

The appraisal of the included papers using the CASP tool revealed generally sound methodological quality across the studies (see S2 Table). Most studies described an appropriate methodology suitable for their stated research aims and objectives. However, a notable deficiency in many studies was observed concerning the declaration of potential biases [62–64]. Additionally, based on the quality judgement, one article [65] appeared to be of relatively low quality compared to the others. Nevertheless, no studies were excluded solely based on quality, as the authors were primarily interested in extracting relevant concepts from all included studies that directly mapped to the systematic review question.

## Study characteristics

The 20 included studies involved participants ranging in age from 10 to 24 years old. Two papers specifically targeted females [64, 75], while the remaining 18 included both male and female. Eight papers focused on youths living with HIV [62, 63, 66, 72–74]. The studies were conducted in various countries, including Ethiopia, Kenya (3), Malawi (1), Nigeria (2), South Africa (6), Uganda (2), and Zambia (5). Seven studies were community-based, 12 were conducted in healthcare facilities, and one was a school-based study.

## Synthesis

The meta-ethnographic synthesis resulted in four key themes encapsulating the breadth of sexual and reproductive health needs and preferences expressed by youths across the included studies; *information needs*, *social needs*, *service needs* and *delivery preferences* (Table 1). These four key themes encompassed multiple nested sub-themes and are represented below with participant quotes.

**Table 1. This is the table showing main themes and sub-themes.**

| Themes | Sub-Themes |
|---|---|
| Information Needs | • Age-appropriate information |
| | • Raising awareness |
| Service Needs | • Health promotion and preventive services |
| | • Treatment services |
| Delivery Preferences | • Provider characteristics |
| | • Logistics of service delivery |
| Social Needs | • Skills acquisition programmes |
| | • Substance use rehabilitation |
| | • Support systems |

## Theme 1—Information needs

This theme emerged across all included studies, highlighting youths' desire for information on safe reproductive and sexual health topics like contraception and condom use. As one participant stated: *"people are no longer given deep information about these things [sexual health], they are only given average information"* [64]. No refuting perspectives arose for this theme, although information preferences differed by gender [66, 67]. Males wanted details on proper condom use—*". . .the need for condoms, you can go to the health facility and they give them to you but there are some people who do not know how to use them"* [68]. Females preferred information on menstruation, relationships, hygiene, safer sex, and abortion to avoid unsafe procedures and consequences: *"They don't get good counselling or advice on how they can protect and take care of the pregnancies. So they end up aborting"* [68]. This theme had two sub-themes: *age-appropriate information* and *raising awareness*.

*Age-Appropriate Information*: Youths expressed needing information tailored to their age covering bodily changes, contraceptives, safer sex, and sexual health issues to meet their needs [67, 69–71]. As one noted: *"I have tried to ask them for more information but they did not give me enough. What they told me was not very useful. . ."* [72]. HIV-positive youths also wanted productive life information regardless of status [72–74], stating: *"We have a right to having a family and have children"* [72]. Some studies revealed misunderstandings around conception and prevention [63, 67, 74]. Youths faced barriers obtaining age-appropriate information, including provider disconnects on suitability—*"sometimes we go to clinics and then they say 'why do you want to know, you are still too young. . .'"* [75]—and discussing sexual health with parents due to stigma [75]. Hence information often came from peers and siblings instead: *"yeah, sisters educate about abortion and they even tell you that abortion is no easy matter because you are then between life and death. They actually teach us about many things"* [75]. Mass media was acknowledged a major source SRH information [75].

*Raising Awareness*: Youths emphasised awareness raising for parents, youths and communities regarding youth needs and preferences to ensure access to information and services [70, 76, 77]. Stigma and misconceptions were cited as barriers: *"Stigma, if you are seen going to the hospital, it's like you're engaging in sex. So, society will have a particular perception of you"* [69]; parents disapproved due to stigma [62, 64]. Parent sensitisation was thus needed: *"Parents do not want to give their children time to access this information, maybe because they feel it is not the right time. . . but generally somebody who is 15 years, that one to me needs lots of counselling and guidance from both home and outside home"* [69]. Community outreach could also raise awareness of available services [68, 70, 76]. If services are perceived as meeting youths' needs, utilisation may increase [14].

## Theme 2: Social needs

Social contexts including the acquisition of skills, substance use rehabilitation, and support systems shaped youth SRH needs. Targeted programmes in these areas were acknowledged to promote healthier sexual and reproductive health behaviours. Three sub-themes encapsulate these social needs: ***Skill acquisition programmes***, ***substance use rehabilitation***, and ***support systems***.

*Skill acquisition programmes*: Youths required skills for healthy relationships and economic productivity [67, 72, 76]. Desired skills included negotiation, decision-making, and refusal skills to avoid unsafe sex practices [67, 72, 76]. As one female noted, such skills could empower responses to unwanted sexual advances: "*If a boy is forcing you. . .what must you do*?" [67]. Vocational skills, such as sewing and catering, were sought to address economic challenges and empower youths, with gender-specific preferences [72]. *". . .one thing that I would really like is being taught life skills because as it is, no one teaches us these things. I would like to learn how to cook, and also tailoring*" [72].

*Substance use rehabilitation*: Rehabilitation was needed to address links between substance use and sexual violence. As youths explained, "*Things that cause rapes are the use of drugs. . . [which] may also cause raping a child*" [69]. lack of job opportunities and peer pressure were identified as substance use triggers-"*Most of the youths don't have what to do—they resort to taking alcohol, opium, cigarettes and marijuana*" [68].

*Support Systems*: Family and peer support encouraged SRH service use [74, 76], "*You need support from people who will understand your condition*" [67]. However, negative parental attitudes hindered communication and support-seeking behaviours: "*When we ask our parents they become aggressive, they shout at us not wanting to talk to us, saying, 'why do want to know about such things'*" [75]. Youths living with HIV express varying attitudes toward family support, highlighting the importance of early parental disclosure for timely treatment: "*The fact my family kept this from me made me very sad; [for a time] I hated my mother*" [74]. Support groups foster positive networks, aiding youths in coping with SRH-related stress, although concerns about disclosure exist among some HIV-positive youths: "*It will hurt if other people know, they will joke about it*" [73].

## Theme 3: Service needs

Youths wanted services directly addressing their sexual and reproductive health issues. The need for a wide range of services was reported due to perceived gaps at local health facilities [65, 67, 69, 71, 77]. This theme had two sub-themes: ***preventive services*** and ***treatment services***.

*Preventive Services*: All studies noted youths wanting health promotion and preventive sexual and reproductive services like HIV testing and counselling, antenatal care, counselling, and contraceptives: "*If we use contraceptives we can have a manageable number of children, rather than having so many that we can't raise them*" [78]. Lacking these, risky alternatives emerged [66–68], example, "*make their own condoms—using bread bags—so they won't be seen trying to get condoms in public*" [67]. Services were seen as only for married couples, not youths: "*I have not received such (family planning services), and I don't think it is made for adolescents. It is only for married couples*" [79].

*Treatment Services*: Youths across all studies wanted STI testing and treatment services for issues like candidiasis, syphilis, and HIV/AIDS [65, 70, 71, 78]: "*. . .when you get infected with diseases like Candida. . .we fall sick all of the time, the trenches here spread diseases due to poor sanitation*" [71]. Reported lack of access may drive this demand: "*She had been to Kasangati health centre and they told her that the drugs are not there. . .*" [68]. One study mentioned

needing timely referrals for serious cases [65]. With unmet treatment needs, youths often relied on peers or unsafe options like herbs or traditional healers [66, 68, 76]: *"When we get problems, sometimes we tell our friends. . .so that is what she is using"* [68]. Access to such services is critical to reduce youth suffering, morbidity and mortality.

### Theme 4- Delivery preferences

The theme indicates youth dissatisfaction with SRHS in healthcare facilities, evident in all included papers, with two subthemes: ***provider characteristics*** and ***logistics of service delivery***.

*Provider characteristics*: Youths prefer providers offering high-quality SRHS, emphasising welcomeness, trustworthiness, respect, confidentiality, friendliness, same-gender providers, and competency [64, 69, 70, 72, 80]. Negative provider attitudes, lack of privacy, and rudeness deter youths from public health facilities [68, 72, 76]. Youths living with HIV requested autonomy in disclosure and challenge providers to respect their confidentiality—*"Everyone was looking at me in a funny way and whispering, 'This is the one who is sick, she has AIDS.' This was very painful for me because I thought that it was my right to disclose to people about my status. I felt that my rights were violated"* [72]. Negative experiences lead youths to favour competent, non-judgmental providers [68, 70].

In some studies, youths express discomfort with providers of the opposite-sex, while others prefer gender-specific providers—*"Sometimes it is difficult or embarrassing to open up to someone of the opposite sex"* [77]. The need for same-gender preference may be related to youth's personal values, which tend to shape the life of an individual. Younger SRH providers were seen as being more equipped to understand and address the SRHS of youths and would relate with the youths in an appropriate manner- *"Younger staff can understand challenges facing adolescents and address our issues as adolescents. . ."* [66]. However, in some other studies youths preferred older SRH providers. Older providers were viewed as experienced and, therefore, competent in the delivery of SRHS—*". . . it is better to get information from experienced adults like your mother or health staff"* [66].

*Logistics of service delivery*: Youths desire dedicated SRHS facilities, adequate staffing, quality medical supplies, free services, short wait times, and youth reward systems [78, 79, 81]. They preferred services tailored to their needs and reported dissatisfaction with existing facilities and claimed they were designed for adults [68, 72, 76]. Youths desired attractive youth-friendly clinics with recreational facilities and enough staff [76]—*". . .I can go there to play, to watch movies, I can be guided, I can be tested"* [76]. Cost and long waits in existing facilities hindered access, prompting a preference for free services and rewards [77, 78]. Financial constraints and poverty further challenged youths' SRHS access to SRH services [71, 72]. Thus, youth-oriented services and understanding providers were lacking, leading youths to relying on social media or traditional healers instead [66, 68, 72, 76, 80].

## Discussion

This review explored the SRHS needs and preferences of youths aged 10–24 in SSA. The synthesis of evidence highlights the importance of targeted information, alignment of SRH services to diverse youth needs, addressing social needs for holistic SRHS, and tailoring delivery preferences to empower and enhance the SRH experiences of this population.

### Youth empowerment through targeted information

A key finding of this review is that youths strongly desire age-appropriate sexual and reproductive health information, but often lack access to reliable sources. Sexual and reproductive health information has demonstrated benefits in preparing youths against risks of unsafe

behaviours [82, 83]. However, youths lack access to age-appropriate information, relying on uncertain social or peer sources instead [4]. For example, in one study [75], youths reported seeking information from older siblings or friends, often receiving inaccurate or incomplete information. Guided, professionally informed education tailored to maturity levels could aid healthy development [84, 85]. The current review supports this finding that SRH education is most effective when it is professionally informed and guided based on the maturity levels and needs of youths. This is because, most youths were ill-prepared on reproductive changes, desiring parental and provider discussions to facilitate access and dispel stigma [68]. Empowering youths with desired knowledge also promote healthy sexuality attitudes [83]. In line with this, our review suggests that guided education may be associated with the development of healthy sexuality perspectives and potentially protective against risky sexual and reproductive behaviours among youths. As youths understand their needs, ensuring youths can access age-appropriate sexual health information is critical to empowering them to understand their bodies and make healthy SRH choices.

## Aligning SRH services to diverse youth needs

Our findings reveal a significant mismatch between available SRH services and the diverse needs of youth in SSA. In the current review, papers identified various youth SRH needs ranging from promotion and preventive to treatment services. Findings demonstrated gaps in aligning care and environments to youth realities. As a result, there is an increased prevalence of unsafe self-help alternatives [68, 86]. For instance, in one study [67], youths reported using makeshift condoms from bread bags due to lack of access to proper contraceptives, highlighting the urgent need for accessible preventive services. This situation, predominant in low- and middle-income countries, where systemic shortcomings, such as social stigma, taboos, and restrictive policies, often hinder the effective positioning of SRH services in relation to youth realities [87–89]. However, in high-income countries, initiatives have aimed at tailoring SRH services to better suit youths, promoting better coordination, and addressing diverse needs [90]. These disparities contribute to adverse sexual and reproductive health outcomes [34, 91]. Thus, emphasising the imperative for increased attention and tailored strategies in these regions to bridge the gap and improve the provision of youth-friendly SRH services [92, 93]. Comprehensive SRH services are embodied rights for youth development [91]. Yet findings showed youths do not seek facility-based SRH services, due to perceived unavailability of services and stigma. Addressing negative social norms through community dialogue and sensitisation could dispel misconceptions and promote access to quality SRH services [94, 95]. Additionally, most youths want quality clinical services from competent providers. Ensuring available, accessible, non-judgmental youth-focused SRH services should be a priority.

## Addressing social needs for holistic SRHS

Our review highlights the interconnectedness of social factors and SRH outcomes among youth in SSA. Acknowledging the intertwined social and economic factors influencing youth SRHS outcomes, skills development programmes that consider gender preferences may facilitate healthy decision-making capacities [5, 72]. Integrated substance abuse and mental health services have also been identified as critical for reducing risky behaviours [96–98]. High-income countries like the UK and US have adopted a more holistic approach, shifting certain centres towards a one-stop model that integrates clinical, social, and educational support [97, 98]. Meanwhile, initiatives in countries such as Mexico, India, and South Africa have incorporated life skills building, education, vocational training, and livelihood support alongside clinical SRH services [99–101]. Such comprehensive models agree with the multifaceted youth

needs and preferences, showing promise for improving access and health outcomes. However, youth-centred comprehensive SRHS remains less uniformly implemented in sub-Saharan Africa [102, 103]. As Hujo and Carter [104] argue, more holistic community-based approaches may be better suited to address the multidimensional youth realities in this region, yet require greater resource mobilisation and political backing for realisation. Additional implementation research could catalyse suitable integrated services for marginalised African youth [102, 104]. In conducting such research, recognising familial and peer support as identified facilitators for SRHS access further emphasises the potential value of social network integration into youth SRHS programmes [105, 106]. Such comprehensive models align with youth preferences and show promise for improving access and health outcomes.

## Tailoring delivery preferences for youths

The review illuminates the significance of provider characteristics (welcomeness, trustworthiness, respect, confidentiality) and logistics of delivery (adequate staffing, quality supplies, free services) in shaping youths' experiences with SRHS [4, 70]. Unfavourable provider attitudes contribute to barriers in accessing SRHS, necessitating training guided by WHO's quality standards [13, 14]. Specifically, WHO's recommendation for youth-friendly health services include accessible locations and hours, lower cost, respectful and confidential care, peer counselling, comprehensive services, provider competencies, youth and community involvement [13, 14]. Preferences for same-sex providers, youth-only centres, and incentives accentuate the importance of recognising and respecting youths' choices for effective SRHS [1, 62, 106]. Recognising and integrating youths' preferences is critical for uptake and sustained engagement with SRHS. The COM-B model identifies capability, opportunity, and motivation as key components that need to be targeted for successful interventions aimed at changing behaviour [107–109]. Addressing these areas in youth-friendly services can directly enable youths to access and engage with sexual and reproductive health services.

## Implications and recommendations for practice

Our findings have several important implications for the delivery of youth SRHS in SSA:

1. There is a clear need for more comprehensive, youth-friendly sexual and reproductive health services tailored to meet the informational and service needs of youth.

2. Stigma and lack of awareness remain significant barriers to youth accessing SRHS.

3. Current healthcare provider practices often do not align with youth preferences and needs.

4. Existing health facilities are not designed with youth needs in mind.

5. There is a strong link between life skills, economic factors, and sexual health outcomes for youth.

   Based on these implications, we recommend the following actions:

1. Develop and implement age-appropriate education programmes, including access to contraception, STI testing and treatment, and counselling.

2. Launch awareness campaigns to reduce stigma around youth sexual health topics in communities and facilitate more open communication with both health providers and parents/families.

3. Provide additional training for healthcare providers on delivering youth-friendly sexual and reproductive health services.

4. Establish dedicated youth health facilities that are appealing, well-staffed, and provide comprehensive, free or low-cost services with minimal wait times.

5. Develop programmes to equip youth with life skills including healthy relationship building, decision making around sexual behaviours, and economic empowerment skills.

6. Establish accessible counselling and rehabilitation services tailored to youth, addressing the links between risky sexual behaviours and issues like substance use.

These recommendations are derived directly from our synthesis of the included studies and aim to address the key needs and preferences identified in our review.

## Strength and limitations

This review boasts several strengths in its systematic approach. We conducted comprehensive searches across multiple databases to identify relevant studies on youth SRHS in SSA. Screening and quality appraisal were meticulously carried out by two independent reviewers, and the utilisation of meta-ethnography methodology adheres to guidance on robust qualitative evidence synthesis. By synthesising insights across primary studies, this review offers qualitative evidence on the delivery situation of youth SRHS in SSA, potentially informing strategic planning to develop safer, more youth-friendly services in this region. However, it is worth noting that as only qualitative research was included, we refrained from making definitive judgments on cause-and-effect relationships. Future mixed-methods reviews could explore provider attitudes and quantitatively assess youth SRHS needs and preferences. Furthermore, while this review specifically represents perspectives within sub-Saharan Africa, broader reviews would facilitate a global comparison of youth needs. Notably, we acknowledge the limitation of not citing grey literature data, which could have provided valuable insights into government and institutional responses to issues such as drug abuse, violence, and HIV, thereby impacting the management of these significant public health challenges..

## Conclusion

This review has indicated that sexual and reproductive healthcare services in SSA are currently not tailored to align with the needs and preferences of youths aged 10–24. This departure from the WHO-recommended delivery gold standard highlights a pressing need to rectify existing gaps in the provision of sexual and reproductive health services. While globally generalisable solutions are unrealistic, a multifaceted approach is required to meet the needs and preferences of youths across information, service, social and delivery realms to ensure access to and well-being of youth sexual and reproductive health. Key youth preferences centre on desiring guided education matched to maturity levels, tailoring of available SRH services to their diverse needs, integration of social dimensions like skills training, and youth-friendly delivery considerations around provider characteristics and atmospheres. Further research is needed to:

1. Develop and evaluate youth-centred models of SRHS delivery that integrate these preferences.

2. Assess the effectiveness of integrated services that combine clinical care with life skills training and economic empowerment programmes.

3. Conduct implementation studies on how to effectively scale up promising interventions in resource-limited settings.

These research directions aim to bridge the gap between youth preferences and current service provision, ultimately improving access and outcomes. Understanding and incorporating

youths' needs and preferences in the planning of sexual and reproductive health services will be imperative to enhancing overall youth welfare in SSA.

## Supporting information

**S1 File. List of countries in sub-Saharan Africa.**
(PDF)

**S2 File. File Search histories.**
(PDF)

**S1 Table. Data extraction summary on youth's needs and preferences regarding SRHS in sub-Saharan Africa.**
(PDF)

**S2 Table. CASP appraisal of included papers on the needs and preferences of youths regarding sexual and reproductive health services in sub-Saharan Africa.**
(PDF)

**S3 Table. List of included and excluded studies.**
(PDF)

**S4 Table. Full data extraction table based on the data extraction tool used.**
(PDF)

**S1 Fig. PICo framework for review question and search terms.**
(PDF)

**S2 Fig. PRISMA flow diagram showing stages of study selection.**
(PDF)

## Acknowledgments

The authors express their gratitude for the support provided during this review. Special thanks to Leah Millard of the University of Manchester, United Kingdom for her dedicated screening of a percentage of titles and abstracts, enhancing the rigour of this review. The authors also extend their appreciation to the participants and researchers whose work contributed to the primary studies included in this review. Their invaluable contributions have enriched the synthesis of evidence presented in this manuscript.

## Author Contributions

**Conceptualization:** Victoria Kalu Uka.

**Formal analysis:** Victoria Kalu Uka.

**Methodology:** Victoria Kalu Uka.

**Supervision:** Helen White, Debbie M. Smith.

**Validation:** Helen White, Debbie M. Smith.

**Writing – original draft:** Victoria Kalu Uka.

**Writing – review & editing:** Victoria Kalu Uka.

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
