## [Decision Letter · Decision Letter 0]

28 May 2024

PONE-D-24-08352The Sexual and Reproductive Health needs and preferences of youths in sub-Saharan Africa: A meta-synthesisPLOS ONE

Dear Dr. Uka,

Thank you for submitting your manuscript to PLOS ONE. After careful consideration, we feel that it has merit but does not fully meet PLOS ONE’s publication criteria as it currently stands. Therefore, we invite you to submit a revised version of the manuscript that addresses the points raised during the review process.

**Please address the reviewers' feedback to improve your manuscript.**

We look forward to receiving your revised manuscript.

Kind regards,

Laura Brunelli, MD, PhD

Academic Editor

PLOS ONE

Journal Requirements:

The authors express their gratitude for the support provided during this review. The first author, Victoria Uka, acknowledges the financial support received from the Tertiary Education Trust Fund, Nigeria, for her doctoral research, which allowed the review process. The funding agency had no involvement in the conception, methods, data synthesis, discussion, manuscript preparation, or the decision to submit the manuscript for publication. Special thanks to Leah Millard of the University of Manchester, United Kingdom for her dedicated screening of a percentage of titles and abstracts, enhancing the rigour of this review. The authors also extend their appreciation to the participants and researchers whose work contributed to the primary studies included in this review. Their invaluable contributions have enriched the synthesis of evidence presented in this manuscript.

Reviewers' comments:

Reviewer's Responses to Questions

**Comments to the Author**

1. Is the manuscript technically sound, and do the data support the conclusions?

Reviewer #1: Yes

Reviewer #2: Partly

2. Has the statistical analysis been performed appropriately and rigorously? 

Reviewer #1: N/A

Reviewer #2: Yes

3. Have the authors made all data underlying the findings in their manuscript fully available?

Reviewer #1: Yes

Reviewer #2: Yes

4. Is the manuscript presented in an intelligible fashion and written in standard English?

Reviewer #1: Yes

Reviewer #2: Yes

5. Review Comments to the Author

**Reviewer #1: **This manuscript presents the results of a metasynthesis review Understanding the sexual and reproductive health (SRH) needs and preferences of youths in sub-Saharan Africa is crucial for informing effective interventions in this critical area. I have included my feedback below for improvement and further modifications.

Title:

• The title effectively communicates the focus of the research, providing clarity on the subject matter. However, it is essential to note that while the authors describe their study as a meta-analysis, no evidence of data pooling or statistical analysis is observed throughout the paper. Instead, the authors synthesize existing evidence, which aligns more closely with a systematic review/ meta-aggregative review rather than a meta-analysis. This distinction should be considered for accuracy and clarity.

Abstract:

• The abstract offers a comprehensive overview of the Review examining the SRH needs and preferences of youths aged 10-24 years in sub-Saharan Africa. The methodology, encompassing database searches and quality appraisal using CASP criteria, is robust and well-documented. Key themes from the synthesis of qualitative research, including information needs, service needs, social needs, and delivery preferences, are effectively summarized.

• However, distinct headings for the background, methods, and results sections facilitate a more straightforward presentation and enhance clarity.

Introduction

• The introduction effectively highlights challenges for SSA youths accessing SRHS and underscores the importance of tailored SRHS based on youth needs.

• It could benefit from a more structured approach to enhance clarity, beginning with a clear statement of the Review's objective and providing a concise background on the topic's significance. This could be followed by systematically discussing barriers to SRHS access and existing recommendations.

• Lastly, a clear transition to the review's focus, exploring the needs and preferences of SSA youths regarding SRHS, would improve coherence. It would also be great if the authors showed us the global data in their research statement.

Method:

• The methods section demonstrates rigorous adherence to PRISMA standards and systematic review protocols, enhancing the study's credibility. Using the PICo framework to structure the review question and search terms ensures clarity and reproducibility. Well-defined eligibility criteria and a robust study selection process contribute to the reliability of the review findings.

• Adopting appropriate quality appraisal tools and a meta-ethnographic approach for data synthesis further strengthens the validity of the study outcomes.

Potential areas that need the Author's attention:

• Search Strategy and Databases: Provide more details on the rationale behind selecting specific search terms and databases. Explain how the search strategy was tailored to ensure comprehensive coverage of relevant literature.

• Study Selection Process: While the inclusion and exclusion criteria are clearly outlined, more transparency in the screening process could be beneficial. Describe any disagreements between reviewers and how they were resolved.

• Clarity on Data Extraction: Details on how qualitative data related to SRHS needs were extracted would enhance transparency.

• Quality Appraisal: Explain why the CASP qualitative checklist was chosen for quality assessment and how it aligns with the Review's objectives. Discuss any challenges encountered during the quality appraisal process.

• Data Synthesis: While the meta-ethnographic approach is described in detail, elaborate on how discrepancies between studies were addressed during synthesis. Provide insights into the interpretation of findings and the development of lines of argument.

Result

Strengths:

• Comprehensive Synthesis: The results comprehensively synthesize the included studies, highlighting key themes and sub-themes related to sexual and reproductive health needs and preferences of youths in sub-Saharan Africa (SSA).

• Clear Presentation: The themes and sub-themes are presented, making it easy to understand the Review's findings.

• Diverse Study Characteristics: The included studies represent a diverse range of participants, including males and females, covering various countries in SSA, ensuring a broad perspective on youth SRH needs.

• Thorough Exploration: Each theme is explored in detail, with participant quotes providing insight into their experiences and perspectives.

Areas for Improvement:

• Inclusion of Refuting Perspectives: While no refuting perspectives arose for the first theme (Information needs), discussing any conflicting findings or perspectives across the other themes would be beneficial to provide a more nuanced understanding of the results.

• Can the authors provide further insights into the representativeness of the included studies in terms of geographic distribution and demographic characteristics of participants?

• Can the authors elaborate on any discrepancies or conflicting findings identified during the synthesis and how they were resolved?

• What implications do the identified themes have for developing and implementing sexual and reproductive health interventions targeting youths in sub-Saharan Africa (If any)?

Discussion

• The Review comprehensively explores the sexual and reproductive health service (SRHS) needs and preferences of youths aged 10-24 in sub-Saharan Africa, providing in-depth analysis and critical reflection on key themes such as youth empowerment, aligning SRH services, addressing social determinants of health, and tailoring delivery preferences. The following may need the author's attention.

• Contextual Considerations: Given the diverse socio-cultural contexts within SSA, it would be beneficial to discuss how factors such as religion, gender norms, and socioeconomic status may influence youth SRHS needs and preferences. Including a nuanced discussion of these contextual factors would enrich the analysis and provide deeper insights into the topic's complexities.

**Reviewer #2:** I would encourage Authors to evaluate the content of grey literature they excluded for insights other than those from reviewing 20 papers. Please properly cite [35]. Also consider dropping the pie graph and use a table instead (you’ll find rationale for this in the attached comments). Avoid redundant information.

6. PLOS authors have the option to publish the peer review history of their article (what does this mean?). If published, this will include your full peer review and any attached files.

Reviewer #1: No

Reviewer #2: **Yes: **Elena Mazzolini

---

## [Author Response · Author response to Decision Letter 0]

25 Jun 2024

Comment on lines 61-67: We have clarified the global statistics and their specific impact on sub-Saharan Africa (SSA). Specifically, we have:

1.Clarified that the 374 million new cases of sexually transmitted infections (STIs) occur globally, with a significant proportion in low- and middle-income countries (LMICs), and emphasised the considerable share borne by youth in SSA.

2. Highlighted that approximately 50% of the 21 million pregnancies among 15-19-year-olds in LMICs occur in SSA, thus providing a clearer understanding of the problem’s magnitude in the study area.

3. Enhanced the overall narrative to better reflect the specific challenges faced in SSA, addressing gender inequality and its impact on sexual and reproductive health.

124: We have revised our manuscript to provide a more detailed justification for the exclusion of grey literature. We acknowledge the potential for publication bias as highlighted by Paez (2017) [35], but we chose to exclude grey literature due to difficulties in comprehensively identifying and accessing these sources. Additionally, the heterogeneous nature and potential lack of rigorous peer review of grey literature pose challenges in assessing the validity and replicability of findings. We ensured the quality and reliability of our sources by focusing on published, peer-reviewed studies and thoroughly searching commercial and open-access databases to ensure a comprehensive collection of relevant studies on the sexual and reproductive health needs and preferences of youths in sub-Saharan Africa. We recognise that excluding grey literature might introduce some bias, but we believe this approach was necessary to maintain the integrity of our meta-synthesis. In future research, we may consider evaluating the content of excluded grey literature for additional insights.

We appreciate the reviewer's comment and would like to clarify the purpose and application of the scoring system. The scoring system was used to enhance the transparency and consistency of our quality assessments. Specifically, assigning scores to questions 1-9 of the CASP checklist facilitated inter-study comparisons and supported our overall quality judgments. For Q10, which is open-ended, we differentiated between valuable and less valuable papers by assigning scores of 2 and 1, respectively. This scoring was based on our qualitative judgment of the paper's value (please refer to the S4 file for revised summary appraisal). The scores helped standardise the evaluation process, ensure consistency, and provide a clear and replicable framework for assessing study quality. Importantly, no studies were excluded based on their scores; all papers were included for their valuable contributions to the meta-ethnographic synthesis. We have revised the quality appraisal section to reflect this explanation.

213-214: Thank you for your insightful comment. We appreciate the opportunity to clarify our approach. Our decision to exclude grey literature was primarily driven by several considerations: 

1. Comprehensive identification and access: Grey literature can be challenging to comprehensively identify and access. Unlike peer-reviewed studies, grey literature is often dispersed across various platforms and lacks standardised indexing, making it difficult to ensure a thorough and systematic search. 

2. Quality and rigorous peer review: One of our main aims was to ensure the scientific quality and reliability of our sources. Published, peer-reviewed studies undergo rigorous review processes that help ensure the validity and replicability of findings. In contrast, grey literature often does not undergo such stringent peer review, posing challenges in assessing its scientific quality. 

3. Heterogeneous nature: Grey literature is highly heterogeneous, varying widely in terms of format, quality, and scope. This heterogeneity complicates the assessment and synthesis of findings, which could affect the overall integrity of our meta-synthesis. 

Regarding the CASP appraisal, while we did not exclude studies solely based on their CASP checklist scores, we did use these scores to inform our understanding of each study's methodological rigor. The inclusion of studies, regardless of their quality scores, was aimed at capturing a wide range of relevant concepts. However, we prioritised published, peer-reviewed studies to maintain a high standard of quality in our review. We acknowledge that excluding grey literature might introduce some bias. In future research, we may consider evaluating the content of excluded grey literature to pick up additional insights that could complement our findings. This approach was necessary to maintain the integrity and reliability of our meta-synthesis. We have documented this rationale in an earlier comment from the reviewers and have ensured a comprehensive collection of relevant studies by thoroughly searching commercial and open-access databases focusing on the sexual and reproductive health needs and preferences of youths in sub-Saharan Africa.

234:Thank you for your insightful comment. We understand that the pie chart may misleadingly suggest an equal distribution among the themes. To address this, we have replaced the pie chart with a table that more accurately represents the main themes and their sub-themes. This format ensures clarity and avoids any misinterpretation. Please find the revised visualisation in Table 1.

372-373: We appreciate the reviewer's feedback. The statement has been revised to reflect the nature of the reviewed papers. The revised sentence now reads: "Our review suggests that guided education may be associated with the development of healthy sexuality perspectives and potentially protective against risky sexual and reproductive behaviours among youths''.

451:We are grateful for the reviewer's valuable feedback. We have revised the section to acknowledge the limitation of not citing grey literature data, which could provide valuable insights into government and institutional responses to significant public health challenges such as drug abuse, violence, and HIV. Thank you for highlighting this important aspect for consideration.

In response to the editor’s comments:

Ethics Statement:

We have added a complete ethics statement in the methods section, indicating that no ethical approval was required for this systematic review as it only involved the analysis of published literature. The statement reads: "As this study is a systematic review of published literature, it did not involve human or animal subjects, and therefore, did not require ethical approval. No primary data collection was conducted, and all data used were from publicly available sources."

Funding Information:

We have moved the funding-related information from the Acknowledgments The updated Funding Statement should now read: "The first author, Victoria Uka, acknowledges the financial support received from the Tertiary Education Trust Fund, Nigeria, for her doctoral research, which facilitated the review process. The funding agency had no involvement in the conception, methods, data synthesis, discussion, manuscript preparation, or the decision to submit the manuscript for publication. The author(s) received no other specific funding for this work."

Acknowledgments Section:

The Acknowledgments section has been revised to remove any funding-related text and now appropriately acknowledges contributions without referencing funding sources.

---

## [Decision Letter · Decision Letter 1]

25 Jul 2024

PONE-D-24-08352R1The Sexual and Reproductive Health needs and preferences of youths in sub-Saharan Africa: A meta-synthesisPLOS ONE

Dear Dr. Uka,

Thank you for submitting your manuscript to PLOS ONE. After careful consideration, we feel that it has merit but does not fully meet PLOS ONE’s publication criteria as it currently stands. Therefore, we invite you to submit a revised version of the manuscript that addresses the points raised during the review process.

**ACADEMIC EDITOR: ************Thank you for all the work done on your manuscript. Some more aspects still need revision, please follow the reviewer’s suggestions and feedback to further improve your work.==============================

We look forward to receiving your revised manuscript.

Kind regards,

Laura Brunelli, MD, PhD

Academic Editor

PLOS ONE

Additional Editor Comments:

Thank you for all the work done on your manuscript. Some more aspects still nedd revision, please follow the reviewer’s suggestions and feedback to further improve your work.

Reviewers' comments:

Reviewer's Responses to Questions

**Comments to the Author**

1. If the authors have adequately addressed your comments raised in a previous round of review and you feel that this manuscript is now acceptable for publication, you may indicate that here to bypass the “Comments to the Author” section, enter your conflict of interest statement in the “Confidential to Editor” section, and submit your "Accept" recommendation.

Reviewer #2: All comments have been addressed

Reviewer #3: (No Response)

2. Is the manuscript technically sound, and do the data support the conclusions?

Reviewer #2: Yes

Reviewer #3: Partly

3. Has the statistical analysis been performed appropriately and rigorously? 

Reviewer #2: N/A

Reviewer #3: N/A

4. Have the authors made all data underlying the findings in their manuscript fully available?

Reviewer #2: Yes

Reviewer #3: Yes

5. Is the manuscript presented in an intelligible fashion and written in standard English?

Reviewer #2: Yes

Reviewer #3: Yes

6. Review Comments to the Author

Reviewer #2: (No Response)

Reviewer #3: Overall: The authors conducted a systematic review sexual and reproductive health service needs among young people aged 10-24 years in sub-Saharan Africa. Findings and discussion are relatively general and, overall, the manuscript would benefit from the follow modifications.

Abstract:

Line 31: First use of CASP should be spelled out

Introduction:

Line 56: consider modifying language – perhaps change “including” to “by” or “due to”

Line 56-57: Suggest modifying or omitting sentence starting “As a result”

Line 59: not necessary to define the acronym SRHS again

Line 74: Purpose of citation [22] is unclear

Methods:

Line 117: Consider replacing “homeless” with “houseless”

Line 129-136:Please resolve redundancy involving “grey literature” – important to be concise

Line 142: Suggest modifying as other sources of information “may lack” or may not have an transparent peer review process

Line 152-153: What was the Kappa level used to confirm agreement/reliability?

Line 186: Best to be consistent with SRHS or SRH, if possible

Line 209: independent researchers or by researchers, independently?

Line 218: It is very odd that a peer-reviewed article was not accessible – it is possible to elaborate further? It also seems inconsistent with the methods where inclusion was considered by title and abstract and full text

Line 230-232: this is redundant – specified in methods

Discussion:

Overall, this section would benefit from additional specific recommendations and findings, where possible.

Conclusion:

Line 489: First sentence is not really a conclusion

Line 500-502: There is quite a lot of literature in this space – are you able to be more specific about the gap and/or the indicated “need” for additional research?

7. PLOS authors have the option to publish the peer review history of their article (what does this mean?). If published, this will include your full peer review and any attached files.

Reviewer #2: No

Reviewer #3: **Yes: **Jake M. Pry

---

## [Author Response · Author response to Decision Letter 1]

20 Aug 2024

Response to Comments

Abstract:

1. Comment: Line 31: First use of CASP should be spelled out

Response: We thank the reviewer for this observation. We have spelled out CASP as ‘’Critical Appraisal Skills Programme (CASP)’’ on its first use in the abstract.

Introduction:

2. Comment: Line 56: consider modifying language – perhaps change ‘’including’’ to ‘’by’’ or ‘’due to’’

Response: We appreciate this suggestion and have changed ‘’including’’ to ‘’due to’’ to improve clarity.

3. Comment: Line 56-57: Suggest modifying or omitting sentence starting ‘’As a result’’

Response: We have modified this sentence to provide more specific information and improve the flow. The new sentence reads: ‘’These barriers contribute to significant disparities in youth SRHS access and utilisation across SSA countries.’’ This revision maintains the concept of inconsistency while adding more depth to our discussion of the challenges in SRHS access.

4. Comment: Line 59: not necessary to define the acronym SRHS again.

Response: We agree and have removed the redefinition of SRHS at this point in the manuscript.

5. Comment: Line 74: Purpose of citation [22] is unclear.

Response: We have reviewed the context of citation [22] and have modified the sentence to clarify its purpose. The revised sentence now reads: ‘’Healthy development is hindered [22] and potentially leads to substance misuse or mental health issues [23]’’. Citation [22] (Hegde, Chandran and Pattnaik, 2022) discusses adolescent development and sexuality from a developmental perspective. It emphasises that adolescence is a critical period for healthy development, including sexual development. The article supports our point that when sexual and reproductive health needs are not met, it can hinder overall healthy development in adolescents. So, to improve clarity, we have ensured that the connection between unmet sexual and reproductive health needs and hindered development is more explicitly stated in the surrounding text. This helps to better contextualise the purpose of citation [22] within our argument. This modification provides a clearer link between the citation and our argument about the importance of addressing adolescents' sexual and reproductive health needs for their overall development.

Methods:

6. Comment: Line 117: Consider replacing ‘’homeless’’ with ‘’houseless’’

Response: We appreciate your suggestion to replace 'homeless' with 'houseless'. However, we have decided to retain the term 'homeless' in our exclusion criteria for several reasons:

• It accurately reflects the terminology used in our original systematic review methodology.

• Changing the term retrospectively could potentially alter the perceived scope of our study.

• To maintain consistency with the broader literature in this field, which predominantly uses the term 'homeless' when discussing excluded populations in similar contexts.

While we acknowledge the evolving nature of terminology in this area, we believe that maintaining our original wording is crucial for the accuracy and reproducibility of our review. Thank you for bringing this important point to our attention.

7. Comment: Line 129-136: Please resolve redundancy involving ‘’grey literature’’- important to be concise.

Response: We had previously expanded this section based on feedback from another reviewer who questioned our exclusion of grey literature. However, we appreciate your emphasis on conciseness, and we have revised this section to provide a clear but more succinct explanation of our approach. We believe this revision maintains the necessary justification while improving the overall readability of our methods section.

8. Comment: Line 142: Suggest modifying as other sources of information ‘’may lack’’ or may not have a transparent peer review process.

Response: We have modified this sentence to read ‘’may lack a transparent peer review process’’ to more accurately reflect the variability in review processes.

9. Comment: Line 152-153: What was the Kappa level used to confirm agreement/reliability?

Response: We have clarified that while we described the process of using Cohen's kappa in the methods section, the specific Kappa values are reported in our results section. We have added a brief mention in the methods that specific Kappa values will be reported in the results.

10. Comment: Line 186: Best to be consistent with SRHS or SRH, if possible.

Response: We have reviewed the manuscript and standardised our use of ‘SRHS’ (Sexual and Reproductive Health Services) when referring to services, retaining ‘SRH’ only when discussing the broader concept of Sexual and Reproductive Health. This distinction allows us to be precise in our language while maintaining consistency. However, we have reviewed the manuscript to ensure consistent use of SRHS where applicable.

11. Comment: Line 209: independent researchers or by researchers, independently?

Response: We have clarified the phrasing about the inter-rater reliability check. The statement now reads: ‘’Prior to finalising the selection process, 10% check of titles and abstracts was conducted on the retrieved articles by two researchers independently (VU and LM) to compute an inter-rater reliability.’’

To address the redundancy you noted (Comment 13: Line 230-232), we have moved this statement to the methods section, replacing a similar but less comprehensive statement that was previously there. This consolidation helps to streamline our methodology description while ensuring all necessary information is retained. These revisions aim to improve clarity and conciseness in our reporting of the review process. We believe these changes address your concerns about both the phrasing and the structure of our methodology description.

12. Comment: Line 218: It is very odd that a peer-reviewed article was not accessible – Is it possible to elaborate further? It also seems inconsistent with the methods where inclusion was considered by title and abstract and full text.

Response: We have elaborated on our attempts to access this specific article and the barriers we encountered. Response: The article in question is: ‘’Reproductive health needs of young persons in markets and motor parks in Southwest Nigeria’’ (https://europepmc.org/article/med/14510129). While we were able to access and screen the abstract of this article during our initial review process, we encountered significant barriers in obtaining the full text:

• The article passed our initial screening based on its title and abstract, which aligned with our inclusion criteria.

• However, when we attempted to access the full text, we found it was not available through open access platforms or our institutional subscriptions.

• The article lacked any contact information for the authors, including email addresses, which prevented us from reaching out directly to request the full text.

• We explored all available avenues to access the full text through our institutional resources, but were unsuccessful in obtaining it.

Given our inability to access the full text despite these efforts, we had to exclude this article. We have rephrased the relevant section concisely to reflect this explanation. We believe this explanation addresses the apparent inconsistency you noted and provides a clearer picture of our comprehensive screening and selection process while maintaining transparency in our methodology.

13. Comment: Line 230-232: this is redundant - specified in methods.

Response: We have removed this redundant statement from the results section as it is already covered in the methods.

Discussion:

14. Comment: Overall, this section would benefit from additional specific recommendations and findings, where possible.

Response: We have addressed this by making the following changes:

• We have restructured the discussion to highlight key findings more explicitly where possible. For the first three subheadings of the discussion (Youth empowerment through targeted information, aligning SRH services to diverse youth needs, and addressing social needs for holistic SRHS), we now begin each section with a clear statement of a main finding. For example: ‘’A key finding of this review is that youths strongly desire age-appropriate sexual and reproductive health information, but often lack access to reliable sources.’’ ‘’Our findings reveal a significant mismatch between available SRH services and the diverse needs of youth in SSA.’’ ‘’Our review highlights the interconnectedness of social factors and SRH outcomes among youth in SSA’’

• We have included more specific examples from the included studies to illustrate our points. For instance, we now mention that ‘’in one study [75], youths reported seeking information from older siblings or friends, often receiving inaccurate or incomplete information,’’ and ‘’in one study [67], youths reported using makeshift condoms from bread bags due to lack of access to proper contraceptives.’’

• We have added a new ‘’Implications and recommendations for practice’’ section, which provides six specific, actionable recommendations based directly on our review findings. We believe these changes provide the additional specific findings and recommendations you suggested, enhancing the practical implications of our review.

Conclusion:

15. Comment: Line 489: First sentence is not really a conclusion

Response: We have revised the opening to more clearly state the main finding of our review. We believe this revision provides a stronger, more conclusive opening that directly states the primary outcome of our study. 

16. Comment: Line 500-502: There is quite a lot of literature in this space – are you able to be more specific about the gap and/or the indicated "need" for additional research?

Response: We have revised the conclusion to provide more specific research directions based on the gaps identified in our review. We now specify three key areas for future research: (1) Developing and evaluating youth-centred SRHS delivery models, (2) Assessing the effectiveness of integrated services, and (3) Conducting implementation studies on scaling up promising interventions.

---

## [Decision Letter · Decision Letter 2]

5 Nov 2024

The Sexual and Reproductive Health needs and preferences of youths in sub-Saharan Africa: A meta-synthesis

PONE-D-24-08352R2

Dear Dr. Uka,

We’re pleased to inform you that your manuscript has been judged scientifically suitable for publication and will be formally accepted for publication once it meets all outstanding technical requirements.

Kind regards,

Laura Brunelli, MD, PhD

Academic Editor

PLOS ONE

Additional Editor Comments (optional):

Reviewers' comments:

Reviewer's Responses to Questions

**Comments to the Author**

1. If the authors have adequately addressed your comments raised in a previous round of review and you feel that this manuscript is now acceptable for publication, you may indicate that here to bypass the “Comments to the Author” section, enter your conflict of interest statement in the “Confidential to Editor” section, and submit your "Accept" recommendation.

Reviewer #2: All comments have been addressed

Reviewer #4: All comments have been addressed

2. Is the manuscript technically sound, and do the data support the conclusions?

Reviewer #2: Yes

Reviewer #4: Yes

3. Has the statistical analysis been performed appropriately and rigorously? 

Reviewer #2: N/A

Reviewer #4: Yes

4. Have the authors made all data underlying the findings in their manuscript fully available?

Reviewer #2: Yes

Reviewer #4: Yes

5. Is the manuscript presented in an intelligible fashion and written in standard English?

Reviewer #2: Yes

Reviewer #4: Yes

6. Review Comments to the Author

Reviewer #2: (No Response)

Reviewer #4: I think the authors have adequately addressed my comments raised in a previous round of review and I feel that this manuscript is now acceptable for publication.To make the manuscript more technically sound, the authors should adhere and incorporate the journal guidelines in their manuscript to make it complete.

7. PLOS authors have the option to publish the peer review history of their article (what does this mean?). If published, this will include your full peer review and any attached files.

Reviewer #2: **Yes: **Elena Mazzolini

Reviewer #4: **Yes: **Alexander Laar
